Association of single nucleotide polymorphisms in ITLN1 gene with ischemic stroke risk in Xi’an population, Shaanxi province

Shi Wenzhen 1
Zhang Qi 1
Lu Ying 2
Liu Jie 1
Ma Xiaojuan 1
Xie Zhen 3
Zhang Gejuan 2
Chang Mingze 2
Tian Ye 2 chhty@sina.com
1 Clinical Medical Research Center, Xi’an Key Laboratory of Cardiovascular and Cerebrovascular Diseases, the Affiliated Hospital of Northwest University, Xi’an No.3 Hospital , Xi’an , China
2 Department of Neurology, the Affiliated Hospital of Northwest University, Xi’an No.3 Hospital , Xi’an , China
3 Xi’an Key Laboratory of Cardiovascular and Cerebrovascular Diseases, The College of Life Sciences and Medicine, Northwest University , Xi’an , China
Mitsouras Katherine
Electronic publication date: 2024 Mar 22
Publication date: 2024
Volume: 12
Electronic Location ID: e16934
Received 2023 Jun 13; Accepted 2024 Jan 22
Copyright: © 2024 Shi et al.
Copyright year: 2024
Copyright holder: Shi et al.
License: This is an open access article distributed under the terms of the Creative Commons Attribution License, which permits unrestricted use, distribution, reproduction and adaptation in any medium and for any purpose provided that it is properly attributed. For attribution, the original author(s), title, publication source (PeerJ) and either DOI or URL of the article must be cited.
License URL: https://creativecommons.org/licenses/by/4.0/

Keywords: Ischemic stroke, Sanger sequencing, ITLN1 gene promoter, Single nucleotide polymorphisms

Funding: Natural Science Foundation of China 82104155 Key Research and Development Program of Shaanxi 2020ZDLSF04-03 and 2021SF-096 Xi’an Science and Technology Planning Project 21YXYY0038 and 21YXYJ0004 Natural Science Basic Research Program of Shaanxi 2022JM-539 and 2022JM-541 This work was supported by the Natural Science Foundation of China (No. 82104155), the Key Research and Development Program of Shaanxi (No. 2020ZDLSF04-03 and 2021SF-096), the Xi’an Science and Technology Planning Project (21YXYY0038 and 21YXYJ0004) and the Natural Science Basic Research Program of Shaanxi (2022JM-539 and 2022JM-541). The funders had no role in study design, data collection and analysis, decision to publish, or preparation of the manuscript.

==============================
Background

Ischemic stroke (IS) is the main cause of death and adult disability. However, the pathogenesis of this complicated disease is unknown. The present study aimed to assess the relationship between ITLN1 single nucleotide polymorphisms (SNPs) and the susceptibility to IS in Xi’an population, Shaanxi province.

Methods

In this study, we designed polymerase chain reaction (PCR) primers located at −3,308 bp upstream of the transcription initiation site within promoter region of the ITLN1 gene. The target fragment was amplified by PCR and identified by agarose gel electrophoresis. Sanger sequencing was then performed in the samples extracted from a cohort comprising 1,272 participants (636 controls and 636 cases), and the obtained sequences were compared with the reference sequences available on the National Center for Biotechnology Information (NCBI) website to detect SNPs in the ITLN1 gene promoter region. Logistic regression analysis was employed to assess the relationship between ITLN1 polymorphisms and IS risk, with adjustments for age and gender. Significant positive results were tested by false-positive report probability (FPRP) and false discovery rate (FDR). The interaction among noteworthy SNPs and their predictive relationship with IS risk were explored using the Multi-Factor Dimensionality Reduction (MDR) software.

Results

The results of Sanger sequencing were compared with the reference sequences on the NCBI website, and we found 14 SNPs in ITLN1 gene promoter satisfied Hardy-Weinberg equilibrium (HWE). Logistic regression analysis showed that ITLN1 was associated with a decreased risk of IS (rs6427553: Homozygous C/C: adjusted OR: 0.69, 95% CI [0.48–0.97]; Log-additive: adjusted OR: 0.83, 95% CI [0.70–0.98]; rs7411035: Homozygous G/G: adjusted OR: 0.66, 95% CI [0.47–0.94]; Dominant G/T-G/G: adjusted OR: 0.78, 95% CI [0.62–0.98]; Log-additive: adjusted OR: 0.81, 95% CI [0.69–0.96]; rs4656958: Heterozygous G/A: adjusted OR: 0.74, 95% CI [0.59–0.94]; Homozygous A/A: adjusted OR: 0.51, 95% CI [0.31–0.84]; Dominant G/A-A/A: adjusted OR: 0.71, 95% CI [0.57–0.89]; Recessive A/A: adjusted OR: 0.59, 95% CI [0.36–0.96]; Log-additive: adjusted OR: 0.73, 95% CI [0.61–0.88]), especially in people aged less than 60 years and males.

Conclusions

In short, our study revealed a correlation between ITLN1 variants (rs6427553, rs7411035 and rs4656958) and IS risk in Xi’an population, Shaanxi province, laying a foundation for ITLN1 gene as a potential biomarker for predicting susceptibility to IS.

Introduction

Stroke, a prevalent and severe manifestation of cerebrovascular disease, can be categorized into ischemic stroke (IS) and hemorrhagic stroke according to the clinical symptoms and signs of brain dysfunction. Approximately 85% of stroke cases are attributed to IS (Della-Morte et al., 2012), making it a major cause of severe disability. Globally, 15 million people suffer from stroke annually, with 5 million succumbing to the disease. In the United States, it is ranked as the fifth deadliest diseases, with stroke statistics from 2009 to 2012 revealing 6.6 million individuals aged over 20 years old affected (Mozaffarian et al., 2016). The prevalence rate of stroke gradually increases with age. In the latest report from the American Heart Association in 2020, the number of stroke cases is projected to reach 7 million (Virani et al., 2020). In China, it is a leading cause of mortality. The occurrence of stroke is associated with various risk factors, including age, gender, overweight, smoking, hypertension, diabetes, diet, lack of exercise and psychological factors (Christensen & Cordonnier, 2021; Drozdz et al., 2021; Harshfield et al., 2021; Park et al., 2023; Rexrode et al., 2022; Sun et al., 2022; Yang et al., 2022; Zhang et al., 2021). Notably, the improvement of people’s living standards has led to changes in diet and lifestyle, resulting in a decrease in the incidence of hemorrhagic stroke. However, the incidence of IS remains high. This imposes significant economic and psychological burdens on both the country and the families of affected patients. Previous studies have highlighted the vital role of candidate genes in IS (Chai, Cao & Lu, 2021; Lin et al., 2022), emphasizing the urgency of screening candidate genes for IS.

Intelectin 1, also known as omentin-1, is an obesity-related factor identified by recent researches. It is mainly expressed in the lung, heart, ovary and placenta (Lesná et al., 2015). Previous studies have shown that the increased expression of omentin-1 can promote angiogenesis and inhibit cell apoptosis (Gu et al., 2017). Recent findings have indicated that omentin-1 expression level is decreased in patients with inflammatory bowel disease, suggesting a potential link with chronic inflammatory diseases (Yin et al., 2015). Additionally, Xu et al. (2018a) have pointed out a negative correlation between elevated omentin-1 expression levels and carotid plaque instability in patients with acute ischemic stroke (AIS), but not in cases of moderate or severe carotid stenosis or occlusion. Consequently, omentin-1 may serve as a biomarker for predicting carotid plaque instability in AIS patients. Xu et al. (2018b) have observed a negative relationship between the higher expression level of omentin-1 in serum and adverse functional outcomes in patients with IS, indicating that omentin-1 may be a biomarker for poor prognosis in AIS patients. Furthermore, in vitro study has found that omentin-1 has a neuroprotective effect on brain hypoxia/reoxygenation injury by activating the GAS6/Axl signaling pathway in neuroblastoma cells (Niu et al., 2021). However, the specific mechanism of omentin-1 in cerebral ischemia requires further investigation.

In the present study, we aimed to assess the relationship between ITLN1 single nucleotide polymorphisms (SNPs) and the susceptibility to IS in Xi’an population, Shaanxi province. Firstly, we completed PCR amplification of ITLN1 gene promoter region, followed by Sanger sequencing, and aligned the obtained sequences with the reference sequences from NCBI to identify SNPs in the ITLN1 gene promoter region. Furthermore, the association of these loci with the risk of IS was analyzed using logistic regression analysis. These findings will lay a foundation for understanding the role of omentin-1 in the pathogenesis of IS.

Materials and Methods

Study subjects

A total of 636 healthy controls (409 males and 227 females) and 636 cases (409 males and 227 females) were recruited from Xi’an No.3 Hospital from 2015 to 2020 (SYXSLL-2019-034). And we mainly collected the basic information (ethnicity, age, gender, history of smoking and alcohol abuse), past medical history (hypertension, atrial fibrillation, diabetes, coronary heart disease) and blood biochemical examination results (lipid and homocysteine levels) of the participants. The inclusion criteria for IS patients were as follows: (1) The age of onset of IS was between 50 and 80 years old (Kumari et al., 2022); (2) IS patients with clinical symptoms of neurological impairment persisting for more than 24 h were confirmed by brain magnetic resonance imaging (MRI)/computed tomography (CT). Some patients with a history of coronary heart disease, cerebral hemorrhage, transient ischemic attack, systemic inflammatory disease or tumors were excluded. Healthy individuals underwent the annual health assessment at the hospital’s physical examination center, and questionnaires were used to ensure that they had no history of cerebrovascular disease, myocardial infarction and so on. The study was conducted in accordance with the ethical principles of the Helsinki Declaration of 1975. Moreover, the study was approved by the ethics committee of Xi’an No.3 Hospital and each patient had written informed consent.

DNA extraction, primer design, DNA amplification and sanger sequencing of PCR products

Genomic DNA was extracted from the blood samples by the GoldMag-Mini Whole Blood Genomic DNA Purification Kit (GoldMag Co. Ltd., Xi’an, China). The concentration and purity were assessed using a Nanodrop 2000 spectrophotometer. PCR primers were designed in the −3,308 bp region upstream of the transcription start site of ITLN1 gene based on the online software-NCBI-primer. The sequences of the forward and reverse primers were synthesized and purified by Tsingke Biotechnology Co., Ltd. and we listed them in Online Resource 1. PCR amplification of target DNA were performed by the following procedures: denaturation at 98 °C for 10 s, annealing at 60 °C for 10 s, and extension at 72 °C for 3 min, with 40 cycles. The temperature was kept at 4 °C to obtain the amplified product. Then, agarose gel electrophoresis was used for identification.

The amplified PCR products of the target fragment were submitted to Tsingke Biotechnology Co., Ltd. for two-way Sanger sequencing. SNPs were identified by comparing with the reference sequences of ITLN1 gene promoter region available from NCBI.

Statistical analyses

In this study, we used G*power 3.1.9.7 software to estimate the sample size of the case and control groups through t-test. The parameter settings were: Tail = 2, Effect size = 0.20, α  = 0.05, Power = 0.945, and Allocation ratio = 1. The Sanger sequencing results of the promoter region of the ITLN1 gene were compared with the reference sequences downloaded from the NCBI website using SnapGene software. Genotype and allele distributions of detected SNPs were compared using Fisher’s exact test. The Hardy-Weinberg equilibrium and genotype distribution in case and control subjects were analyzed using the χ2 test. With the SPSS software version 24, odds ratios (adjusted OR) and 95% confidence intervals (CIs) were calculated by logistic regression analysis to assess the association between omentin-1 polymorphisms and IS risk using different genetic models (codominant, dominant, recessive and log-additive). Suppose that A was the dominant allele and B was the recessive allele. In the codominant model, individuals with different genotypes (eg.,: AA, AB, BB) exhibited different phenotypic characteristics. In the dominant model, individuals possessing at least one risk allele (either heterozygous or homozygous) were contrasted with those lacking the risk allele (homozygous). In the recessive model, the phenotype was manifested exclusively in individuals homozygous for the risk allele, while heterozygotes exhibited the same phenotype as homozygotes without the risk allele. In the log-additive model, each additional risk allele intensified the association with the phenotype in a consistent proportion, signifying an additive increase in risk. Significant findings were subjected to false-positive report probability (FPRP) at a prior probability level of “0.25, 0.1, 0.01, 0.001, 0.0001” (adjusted OR = 1.5), and false discovery rate (FDR) was utilized to rule out false positives. Multi-factor dimensionality reduction (MDR) software was employed to examine the interaction (synergy or antagonism) among significant SNPs to predict its relationship with IS risk. p < 0.05 indicated statistical significance.

Results

The information of study subjects and ITLN1 variants

In total, 636 controls (409 males and 227 females) and 636 cases (409 males and 227 females) were recruited. The two groups were equal in size, with an even distribution of males and females. The mean ages of the two groups were 56.04 ± 0.36 years old and 55.51 ± 0.28 years old, respectively, which showed no significant difference in age between the case and control groups (p = 0.254).

After comparing the Sanger sequencing results of the ITLN1 gene promoter region with the reference sequences downloaded from NCBI website, 18 SNPs (rs77824633, rs79209815, rs112656766, rs12094513, rs12094649, rs6427553, rs12094703, rs12091185, rs7411035, rs74547401, rs4656958, rs58170012, rs111321091, rs80262455, rs60429964, rs12091040, rs58615002 and rs60721814) were identified in this study. rs60429964, rs12091040, rs58615002 and rs60721814 did not conform to HWE, so they were excluded from further analysis. The basic information about the remaining 14 selected SNPs in ITLN1 gene, including the SNP-ID, chromosome, position, consequence, alleles, minor allele frequency (MAF) in cases and controls, as well as HWE p-value, was presented in Table 1. As shown in the allele model, people with rs6427553-C, rs7411035-G and rs4656958-A had a lower risk of IS (all p < 0.05). The functions of the selected SNPs predicted by RegulomeDB and HaploReg v4.2 were shown in Online Resource 2.

Table 1 The basic information of selected variants in ITLN1.

SNP-ID	Chr: Position	Consequence	Allele	MAF	HWE	OR (95%CI)	p-value	FDR p-value	
Case	Control	p-value	
rs77824633	1: 160,884,959	Intron	C/T	0.135	0.130	0.384	1.05 [0.83–1.32]	0.702	1.229	
rs79209815	1: 160,885,137	5′ UTR	C/T	0.087	0.080	1.000	1.11 [0.84–1.47]	0.481	1.346	
rs112656766	1: 160,885,146	5′ UTR	C/T	0.047	0.048	1.000	0.99 [0.68–1.42]	0.940	1.196	
rs12094513	1: 160,885,624	2KB Upstream	A/G	0.133	0.124	0.464	1.08 [0.85–1.36]	0.533	1.243	
rs12094649	1: 160,885,921	2KB Upstream	A/G	0.133	0.126	0.280	1.07 [0.85–1.35]	0.574	1.147	
rs6427553	1: 160,886,023	2KB Upstream	T/C	0.364	0.406	0.323	0.84 [0.72–0.99]	0.032	0.151	
rs12094703	1: 160,886,043	2KB Upstream	A/G	0.133	0.126	0.280	1.07 [0.85–1.35]	0.574	1.147	
rs12091185	1: 160,886,188	2KB Upstream	C/T	0.131	0.127	0.206	1.04 [0.83–1.31]	0.732	1.025	
rs7411035	1: 160,886,286	2KB Upstream	T/G	0.369	0.416	0.463	0.82 [0.70–0.97]	0.017	0.119	
rs74547401	1: 160,887,040	2KB Upstream	T/C	0.134	0.123	0.584	1.10 [0.87–1.38]	0.440	1.538	
rs4656958	1: 160,887,174	2KB Upstream	G/A	0.243	0.297	0.058	0.76 [0.64–0.90]	0.002	0.029	
rs58170012	1: 160,887,586	3KB Upstream	T/C	0.135	0.131	0.483	1.04 [0.83–1.31]	0.723	1.124	
rs111321091	1: 160,887,644	3KB Upstream	G/A	0.135	0.131	0.483	1.04 [0.83–1.31]	0.723	1.124	
rs80262455	1: 160,887,880	3KB Upstream	G/A	0.135	0.131	0.483	1.04 [0.83–1.31]	0.723	1.124	
Note:

SNP, single nucleotide polymorphism; MAF, minor allele frequency; HWE, Hardy-Weinberg equilibrium. p-value was calculated by Person’s chi-square test and p < 0.05 is indicated in bold.

The association between ITLN1 variants and IS risk

We further explored the role of these loci in IS risk in both IS patients and healthy controls under four genetic models, as shown in Table 2. rs6427553, rs7411035 and rs4656958 were associated with a decreased risk of IS (rs6427553: homozygous C/T: adjusted OR: 0.69; log-additive: adjusted OR: 0.83; rs7411035: homozygous G/G: adjusted OR: 0.66; dominant G/G-G/T: adjusted OR: 0.78; log-additive: adjusted OR: 0.75; rs4656958: heterozygous G/A: adjusted OR: 0.74; homozygous A/A: adjusted OR: 0.51; dominant A/A-A/A: adjusted OR: 0.71; recessive A/A: adjusted OR: 0.59; log-additive: adjusted OR: 0.73). The significant results were displayed in the forest map (Fig. 1). Subsequently, we used FPRP to screen for false positives in significant results (rs6427553: C vs T: Power = 0.997, FPRP value = 0.101; C/C vs T/T: Power = 0.578, FPRP value = 0.145; rs7411035: G vs T: Power = 0.983, FPRP value = 0.059, 0.157; G/G vs T/T: Power = 0.478, FPRP value = 0.118; G/G vs G/T-T/T: Power = 0.911, FPRP value = 0.098; rs4656958: A vs G: Power = 0.923, FPRP value = 0.009, 0.027; A/A vs A/G: Power = 0.804, FPRP value = 0.048, 0.132; A/A vs G/G: Power = 0.146, FPRP value = 0.143; A/A vs A/G-G/G: Power = 0.708, FPRP value = 0.112, 0.036) at the prior probabilities of 0.25 and 0.1 (Table 3).

Table 2 The association between ITLN1 polymorphisms and ischemic stroke risk.

SNP-ID	Model	Genotype	Frequency	With adjustment	FDR p-value	
Case	Control	OR (95% CI)	p-value	
rs6427553	Codominant	T/T	250	218	1			
		C/T	305	319	0.83 [0.65–1.05]	0.125	0.583	
		C/C	78	98	0.69 [0.48–0.97]	0.035	0.165	
	Dominant	T/T	250	218	1			
		C/T- C/C	383	417	0.80 [0.63–1.00]	0.050	0.234	
	Recessive	T/T -C/T	555	537	1			
		C/C	78	98	0.76 [0.55–1.06]	0.103	0.359	
	Log-additive	–	–	–	0.83 [0.70–0.98]	0.025	0.117	
rs7411035	Codominant	T/T	247	212	1			
		G/T	303	318	0.81 [0.64–1.04]	0.094	0.655	
		G/G	82	105	0.66 [0.47–0.94]	0.020	0.140	
	Dominant	T/T	247	212	1			
		G/G-G/T	385	423	0.78 [0.62–0.98]	0.031	0.218	
	Recessive	G/T-T/T	550	530	1			
		G/G	82	105	0.75 [0.55–1.03]	0.071	0.500	
	Log-additive	–	–	–	0.81 [0.69–0.96]	0.014	0.097	
rs4656958	Codominant	G/G	355	304	1			
		G/A	253	286	0.74 [0.59–0.94]	0.012	0.162	
		A/A	28	46	0.51 [0.31–0.84]	0.009	0.121	
	Dominant	G/G	355	304	1			
		A/A-G/A	281	332	0.71 [0.57–0.89]	0.003	0.039	
	Recessive	G/A-G/G	608	590	1			
		A/A	28	46	0.59 [0.36–0.96]	0.032	0.453	
	Log-additive	–	–	–	0.73 [0.61–0.88]	0.001	0.013	
Note:

SNP, single nucleotide polymorphism; OR, odds ratio; 95% CI, 95% confidence interval. p-value was calculated by logistic regression analysis with adjustments for age and gender. Bold values indicated that the p-value was statistically significant.

Figure 1 The significant association between rs6427553, rs7411035 and rs4656958 and IS risk.

Table 3 Results of FPRP analysis for significant findings.

Model	OR (95%CI)	Power	Prior probability	
0.25	0.1	0.01	0.001	0.0001	
rs6427553								
T vs C	0.84 [0.72–0.99]	0.997	0.101	0.253	0.788	0.974	0.997	
T/T vs C/C	0.69 [0.48–0.97]	0.578	0.145	0.337	0.849	0.983	0.998	
rs7411035								
T vs G	0.82 [0.70–0.97]	0.983	0.059	0.157	0.673	0.954	0.995	
T/T vs G/G	0.66 [0.47–0.94]	0.478	0.118	0.286	0.815	0.978	0.998	
T/T vs G/G-G/T	0.78 [0.62–0.98]	0.911	0.098	0.245	0.781	0.973	0.997	
rs4656958								
G vs A	0.76 [0.64–0.91]	0.923	0.009	0.027	0.233	0.754	0.968	
G/G vs A/G	0.74 [0.59–0.94]	0.804	0.048	0.132	0.627	0.944	0.994	
G/G vs A/A	0.51 [0.31–0.84]	0.146	0.143	0.334	0.847	0.982	0.998	
G/G vs A/A-A/G	0.71 [0.57–0.89]	0.708	0.012	0.036	0.294	0.807	0.977	
Note:

The level of false-positive report probability (FPRP) threshold was set at 0.2 and noteworthy findings are presented and FPRP value < 0.2 is indicated in bold.

Association between ITLN1 polymorphisms and IS risk stratified by age and gender

Age- and gender-stratified analyses were performed to assess the association between ITLN1 polymorphisms and IS risk (Tables 4 and 5). As for age stratification analysis, we used age 60 as the cut-off point and divided the control and IS groups into those who were less than 60 years and those who were greater than or equal to 60 years. In Table 4, in people aged less than 60 years, rs6427553, rs7411035 and rs4656958 were studied to be associated with a decreased risk of IS in the allele (rs6427553: adjusted OR: 0.74; rs7411035: adjusted OR: 0.72; rs4656958: adjusted OR: 0.60), codominant (rs6427553 homozygous T/T: adjusted OR: 0.57; rs7411035 homozygous T/T: adjusted OR: 0.53; rs4656958 heterozygous G/A: adjusted OR: 0.65; homozygous G/G: adjusted OR: 0.23), dominant (rs6427553 C/T-T/T: adjusted OR: 0.74; rs7411035 G/T-T/T: adjusted OR: 0.72; rs4656958 G/A-G/G: adjusted OR: 0.59), recessive (rs6427553 T/T: adjusted OR: 0.65; rs7411035 T/T: adjusted OR: 0.61; rs4656958 G/G: adjusted OR: 0.29) and log-additive (rs6427553: adjusted OR: 0.77; rs7411035: adjusted OR: 0.74; rs4656958: adjusted OR: 0.58) models. Also, in males, rs6427553, rs7411035 and rs4656958 were related to a decreased risk of IS in the allele (rs6427553-C: adjusted OR: 0.76; rs7411035-G: adjusted OR: 0.75; rs4656958-A: adjusted OR: 0.60), codominant (rs6427553 C/T: adjusted OR: 0.70; rs6427553 C/C: adjusted OR: 0.52; rs7411035 G/T: adjusted OR: 0.70; rs7411035 G/G: adjusted OR: 0.50; rs4656958 G/A: adjusted OR: 0.62; rs4656958 A/A: adjusted OR: 0.08), dominant (rs6427553 C/T-T/T: adjusted OR: 0.66; rs7411035 G/G-G/T: adjusted OR: 0.66; rs4656958 A/A-G/A: adjusted OR: 0.55), recessive (rs6427553 C/C: adjusted OR: 0.64; rs7411035 G/G: adjusted OR: 0.70; rs4656958 A/A: adjusted OR: 0.10), and log-additive (rs6427553: adjusted OR: 0.71; rs7411035: adjusted OR: 0.70; rs4656958: adjusted OR: 0.52) models (Table 5). However, no significant association of rs6427553, rs7411035 and rs4656958 with IS risk was found in subjects aged ≥ 60 years and females.

Table 4 Association between ITLN1 polymorphisms and ischemic stroke risk in the case-control groups stratified by age.

SNP	Model	Genotype	Frequency	≥60	Frequency	<60	
Case	Control	OR (95% CI)	p-value	FDR p-value	Case	Control	OR (95% CI)	p-value	FDR p-value	
rs6427553	Allele	T	244	290	1			561	465	1			
		C	148	166	0.84 [0.72–0.99]	0.685	1.198	313	349	0.74 [0.61–0.90]	0.003	0.014	
	Codominant	T/T	75	86	1			175	132	1			
		C/T	94	118	0.78 [0.48–1.28]	0.322	1.500	211	201	0.80 [0.59–1.10]	0.169	0.295	
		C/C	27	24	1.39 [0.65–3.01]	0.405	1.134	51	74	0.57 [0.37–0.89]	0.013	0.059	
	Dominant	T/T	75	86	1			175	132	1			
		C/T-T/T	121	142	0.87 [0.54–1.39]	0.550	1.539	262	275	0.74 [0.55–0.99]	0.048	0.223	
	Recessive	C/C-C/T	169	204	1			386	333	1			
		C/C	27	24	1.61 [0.78–3.30]	0.195	1.366	51	74	0.65 [0.43–0.97]	0.034	0.159	
	Log-additive	–	–	–	1.04 [0.73–1.47]	0.845	1.972	–	–	0.77 [0.62–0.94]	0.013	0.059	
rs7411035	Allele	T	238	283	1			559	459	1			
		G	154	173	0.82 [0.70–0.97]	0.688	1.070	313	355	0.72 [0.60–0.88]	0.001	0.008	
	Codominant	T/T	73	83	1			174	129	1			
		G/T	92	117	0.75 [0.46–1.23]	0.250	3.500	211	201	0.79 [0.58–1.07]	0.130	0.259	
		G/G	31	28	1.04 [0.50–2.17]	0.917	1.604	51	77	0.53 [0.34–0.82]	0.004	0.031	
	Dominant	T/T	73	83	1			174	129	1			
		G/G-G/T	123	145	0.80 [0.50–1.29]	0.356	4.983	262	278	0.72 [0.53–0.96]	0.027	0.186	
	Recessive	G/T-T/T	165	200	1			385	330	1			
		G/G	31	28	1.24 [0.63–2.43]	0.534	1.246	51	77	0.61 [0.41–0.91]	0.015	0.102	
	Log-additive	–	–	–	0.94 [0.67–1.33]	0.727	2.545	–	–	0.74 [0.60–0.91]	0.005	0.032	
rs4656958	Allele	G	285	348	1			678	546	1			
		A	109	110	0.76 [0.64–0.91]	0.225	3.143	200	268	0.60 [0.48–0.75]	0.000	0.000	
	Codominant	G/G	107	132	1			248	172	1			
		G/A	71	84	0.76 [0.46–1.23]	0.255	1.784	182	202	0.65 [0.49–0.87]	0.003	0.046	
		A/A	19	13	1.39 [0.54–3.58]	0.500	0.999	9	33	0.23 [0.11–0.51]	0.000	0.003	
	Dominant	G/G	107	132	1			248	172	1			
		A/A-G/A	90	97	0.83 [0.52–1.31]	0.423	2.960	191	235	0.59 [0.45–0.79]	0.000	0.004	
	Recessive	G/A-G/G	178	216	1			430	374	1			
		A/A	19	13	1.55 [0.61–3.90]	0.358	1.253	9	33	0.29 [0.13–0.62]	0.001	0.020	
	Log-additive	–	–	–	0.95 [0.66–1.37]	0.786	2.202	–	–	0.58 [0.46–0.75]	0.000	0.000	
Note:

SNP, single nucleotide polymorphism; OR, odds ratio; 95% CI, 95% confidence interval. Bold values indicated that the p-value was statistically significant.

Table 5 Association between ITLN1 polymorphisms and ischemic stroke risk in the case-control groups stratified by gender.

SNP	Model	Genotype	Frequency	Female	Frequency	Male	
Case	Control	OR (95% CI)	p-value	FDR p-value	Case	Control	OR (95% CI)	p-value	FDR p-value	
rs6427553	Allele	T	244	290	1			263	266	1			
		C	148	166	1.01 [0.77–1.31]	0.965	1.350	187	188	0.76 [0.62–0.93]	0.007	0.030	
	Codominant	T/T	75	86	1			78	82	1			
		C/T	94	118	1.14 [0.75–1.74]	0.532	3.723	107	102	0.70 [0.52–0.95]	0.020	0.142	
		C/C	27	24	1.01 [0.59–1.73]	0.964	1.500	40	43	0.52 [0.32–0.83]	0.007	0.032	
	Dominant	T/T	75	86	1			78	82	1			
		C/T-T/T	121	142	1.10 [0.75–1.63]	0.62	1.240	147	145	0.66 [0.50–0.89]	0.005	0.025	
	Recessive	C/C-C/T	169	204	1			185	184	1			
		C/C	27	24	0.94 [0.58–1.52]	0.797	1.395	40	43	0.64 [0.41–0.99]	0.045	0.211	
	Log-additive	–	–	–	1.03 [0.79–1.33]	0.847	1.318	–	–	0.71 [0.57–0.89]	0.002	0.010	
rs7411035	Allele	T	238	283	1			258	257	1			
		G	154	173	0.97 [0.75–1.26]	0.826	1.445	192	197	0.75 [0.61–0.91]	0.005	0.032	
	Codominant	T/T	73	83	1			77	78	1			
		G/T	92	117	1.08 [0.71–1.65]	0.725	2.029	104	101	0.70 [0.52–0.95]	0.022	0.102	
		G/G	31	28	0.92 [0.55–1.56]	0.763	1.335	44	48	0.50 [0.31–0.80]	0.004	0.027	
	Dominant	T/T	73	83	1			77	78	1			
		G/G-G/T	123	145	1.03 [0.69–1.52]	0.891	1.247	148	149	0.66 [0.49–0.88]	0.005	0.034	
	Recessive	G/T-T/T	165	200	1			181	179	1			
		G/G	31	28	0.88 [0.56–1.41]	0.602	1.204	44	48	0.61 [0.39–0.95]	0.028	0.193	
	Log-additive	–	–	–	0.97 [0.75–1.26]	0.842	1.473	–	–	0.70 [0.57–0.87]	0.001	0.010	
rs4656958	Allele	G	285	348	1			308	315	1			
		A	109	110	1.07 [0.81–1.42]	0.617	1.439	146	139	0.60 [0.48–0.76]	0.000	0.000	
	Codominant	G/G	107	132	1			106	108	1			
		G/A	71	84	1.07 [0.72–1.59]	0.745	1.739	96	99	0.62 [0.46–0.82]	0.001	0.014	
		A/A	19	13	1.25 [0.65–2.42]	0.503	1.175	25	20	0.08 [0.02–0.28]	0.000	0.001	
	Dominant	G/G	107	132	1			106	108	1			
		A/A-G/A	90	97	1.10 [0.76–1.60]	0.617	1.441	121	119	0.55 [0.42–0.73]	0.000	0.001	
	Recessive	G/A-G/G	178	216	1			202	207	1			
		A/A	19	13	1.21 [0.65–2.28]	0.547	1.531	25	20	0.10 [0.03–0.34]	0.000	0.003	
	Log-additive	–	–	–	1.10 [0.83–1.46]	0.514	1.200	–	–	0.52 [0.40–0.67]	0.000	0.000	
Note:

SNP, single nucleotide polymorphism; OR, odds ratio; 95% CI, 95% confidence interval. Bold values indicated that the p-value was statistically significant.

SNP-SNP interaction analyzed by the MDR software

Subsequently, we examined the impact of potential SNP-SNP interactions on IS risk using the MDR software (Table 6). The three-locus model containing ITLN1 variants (rs6427553, rs7411035 and rs4656958) was considered as the best model for predicting IS risk (cross-validation consistency: 10/10, testing balanced accuracy: 54.3%, adjusted OR: 1.54, 95% CI [1.23–1.92], p < 0.000). We will also present the circle graph (Fig. 2) and the interaction among rs6427553, rs7411035 and rs4656958 was antagonistic, with the information gain values of −0.25%, −0.45%, and −0.30%, respectively.

Table 6 SNP-SNP interaction models of candidate SNPs analyzed by the MDR method.

Model	Training Bal. Acc.	Testing Bal. Acc.	CVC	OR (95% CI)	p	
rs4656958	0.540	0.540	10/10	1.38 [1.11–1.72]	0.004	
rs7411035, rs4656958	0.551	0.537	10/10	1.51 [1.20–1.89]	p < 0.000	
rs6427553, rs7411035, rs4656958	0.553	0.543	10/10	1.54 [1.23–1.92]	p < 0.000	
Note:

MDR, multi-factor dimensionality reduction; Bal. Acc, balanced accuracy; CVC, cross-validation consistency; OR, odds ratio; 95% CI, 95% confidence interval. Bold values indicate that the p-value was statistically significant. p-values were calculated using χ2 tests.

Figure 2 In the circle graph, the interaction between rs6427553, rs7411035 and rs4656958 was antagonistic, with the information gain value of −0.25%, −0.45%, and −0.30%.

Discussion

Ischemic stroke is one of the most important causes of morbidity and mortality, resulting from an intricate interplay of factors, and the importance of genetic factors cannot be ignored. In this study, ITLN1 (rs6427553, rs7411035 and rs4656958) was investigated to be associated with a decreased risk of IS, particularly in people aged less than 60 years and males. In other words, these loci may exert a protective effect against IS, thereby providing a theoretical basis for the potential use of this gene as a biomarker for IS.

Omentin-1 is a novel adipokine expressed in the stromal vascular cells of visceral adipose tissue. It has been reported that omentin-1 is inversely related to cardiovascular and cerebrovascular diseases, including coronary heart disease, endothelial dysfunction (Liu et al., 2020) and IS (Lin et al., 2021). In addition, there is growing recognition of omentin-1’s potential as a promising biomarker for IS. A diminished level of omentin-1 appears to correlate with an elevated risk of developing IS. Conversely, an increased level of omentin-1 is associated with a reduced incidence of IS (Lin et al., 2021). This suggests that omentin-1 levels may exhibit a negative correlation with poor prognosis of IS.

ITLN1 variants have been implicated in various chronic diseases, such as human kidney stone disease (Pungsrinont et al., 2021) and cardiovascular disease (Al-Barqaawi et al., 2022; Vimaleswaran et al., 2021; Zhang et al., 2020). Nazar, Zehra & Azhar (2017) have discovered an association of ITLN1 Val109Asp with coronary heart disease in the Pakistani population (Nazar, Zehra & Azhar, 2017) and the Turkish population (Güçlü-Geyik et al., 2022). The link between ITLN1 SNPs and cardiometabolic disease risk in Asian Indians has been evidenced by Vimaleswaran et al. (2021). Al-Barqaawi et al. (2022) have investigated the effect of ITLN1 SNPs (rs2274907 and rs2274908) on coronary artery disease risk in Iraqi individuals. In a study reported by Zhang et al. (2020), it was found that variants of ITLN1 gene may lead to several clinical phenotypes of systemic lupus erythematosus in the Chinese people. Nevertheless, as of now, there is no reported the relationship between ITLN1 variants and IS risk in the Chinese population.

Previous studies have indicated that the incidence of stroke rises with age, with a higher prevalence among the elderly population (Rexrode et al., 2022). Gender is also identified as a key factor influencing the occurrence of stroke (Rexrode et al., 2022). In our study, a stratified analysis by age revealed that rs6427553, rs7411035 and rs4656958 were still associated with a decreased risk of IS in people aged less than 60 years. However, no such association was found in individuals aged over 60 years. Furthermore, our findings showed that rs6427553, rs7411035 and rs4656958 were related to a decreased risk of IS in males, but not in females. Consequently, our research suggested that the influence of ITLN1 polymorphisms (rs6427553, rs7411035 and rs4656958) on the risk of IS are contingent on age- and gender-dependent. Nevertheless, the underlying mechanisms of these age- and gender-dependent effects remain unclear.

After comparing the results of Sanger sequencing with the reference sequences from the NCBI website, we identified 18 SNPs in the promoter region of ITLN1 gene. Based on the logistic regression analysis, we firstly elucidated a reduced correlation between ITLN1 variants (rs6427553, rs7411035 and rs4656958) and IS risk in Xi’an population, Shaanxi province, laying a foundation for understanding the role of ITLN1 gene in IS. However, this study has some shortcomings. First of all, the selected samples are mainly from a single hospital, and the sample size is not large. In follow-up studies, we intend to expand our sample pool by encompassing diverse hospitals, populations, and ethnicities to validate and generalize our findings. Secondly, we investigated the function of the selected SNPs predicted by RegulomeDB (http://www.regulomedb.org/) and HaploReg v4.2 (https://pubs.broadinstitute.org/mammals/haploreg/haploreg.php), as shown in Online Resource 2. Despite this, the role of omentin-1 and its sites in IS remains unclear. Molecular experiments are needed to provide a theoretical basis for comprehending the mechanism of omentin-1 in the onset of IS.

Conclusion

In the study, we have observed a correlation between ITLN1 variants (rs6427553, rs7411035 and rs4656958) and a reduced risk of IS in the Xi’an population, Shaanxi province.

Supplemental Information

Supplemental Information 1 Supplemental Tables.

Supplemental Information 2 Data.

We thank all authors for their contributions and supports. We are also grateful to all subjects for providing blood samples.

Additional Information and Declarations

Competing Interests

Author Contributions

Human Ethics

Data Availability

The authors declare that they have no competing interests.

Wenzhen Shi conceived and designed the experiments, prepared figures and/or tables, and approved the final draft.

Qi Zhang performed the experiments, prepared figures and/or tables, and approved the final draft.

Ying Lu performed the experiments, prepared figures and/or tables, and approved the final draft.

Jie Liu analyzed the data, prepared figures and/or tables, and approved the final draft.

Xiaojuan Ma analyzed the data, prepared figures and/or tables, and approved the final draft.

Zhen Xie analyzed the data, prepared figures and/or tables, and approved the final draft.

Gejuan Zhang analyzed the data, authored or reviewed drafts of the article, and approved the final draft.

Mingze Chang analyzed the data, authored or reviewed drafts of the article, and approved the final draft.

Ye Tian conceived and designed the experiments, authored or reviewed drafts of the article, and approved the final draft.

The following information was supplied relating to ethical approvals (i.e., approving body and any reference numbers):

Xi’an No.3 Hospital (SYXSLL-2019034).

The following information was supplied regarding data availability:

The raw data are available in the Supplemental File.

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
