# Peer review of "Association of single nucleotide polymorphisms in ITLN1 gene with ischemic stroke risk in Xi’an population, Shaanxi province"

_PeerJ, doi:10.7717/peerj.16934_

## Round 0.1 · original submission · Major Revisions

Your manuscript was considered interesting by the reviewers however they had a number of significant concerns that need to be addressed. First, they suggested that you provide more background in your introduction, and specifically more background on the ITLN1 gene, use more current references, and that the abstract be reorganized with more information in the results section. The reviewers also wanted to know the rationale for selecting subjects 40 years old or older and for excluding subjects with coronary heart disease and systemic inflammatory diseases. An additional issue the reviewers wanted you to address is regarding how you adjusted for confounders in your logistic regression analysis. They would also like to see an in silico analysis focusing on the functional effect of the variants you identified as associated with ischemic stroke. Lastly, the reviewers suggested you include a section discussing the limitations of your study, such as potential sources of bias and whether your results are applicable to the entire population of China. One of the reviewers kindly provided an annotated copy of your manuscript.

Please, submit a detailed rebuttal which shows where and how you have taken all comments and suggestions into consideration. If you do not agree with some of the reviewers’ comments or suggestions, please explain why. Your rebuttal will be critical in making a final decision on your manuscript. Please, note also that your revised version may enter a new round of review by the same or by different reviewers. Therefore, I cannot guarantee that your manuscript will eventually be accepted.

**Language Note:** The review process has identified that the English language must be improved. PeerJ can provide language editing services - please contact us at [email protected] for pricing (be sure to provide your manuscript number and title). Alternatively, you should make your own arrangements to improve the language quality and provide details in your response letter. – PeerJ Staff

Reviewer 1 ·

Basic reporting

English language needs to be checked and polished.

Experimental design

Methods: why did the authors select the IS patients older than 40 years? The authors focus on IS, so I can understand the authors exclude patients with cerebral hemorrhage were excluded. Why did they exclude patients with coronary heart disease or systemic inflammatory diseases. Any references?

Validity of the findings

no comment

Additional comments

In this case-control study, the authors investigated potential associations between intelectin 1 (ITLN1) genetic variations and susceptibility to ischemic stroke (IS) in a Chinese Han population. Outcomes based on 636 patients and 636 controls demonstrated that rs6427553, rs7411035 and rs4656958 may associate with reduced risks of IS development in this cohort. This study uncovers the pathogenesis of IS from perspective of SNPs and is novel and interesting. Several issues should be addressed or revised.
1. Abstract: Please reorganize this part, especially for the results section. The outcomes should be described in detail.
2. Methods: why did the authors select the IS patients older than 40 years? The authors focus on IS, so I can understand the authors exclude patients with cerebral hemorrhage were excluded. Why did they exclude patients with coronary heart disease or systemic inflammatory diseases. Any references?
3. Results: I would suggest using table instead of describing outcomes to display the results.
4. Discussion: Any comments on the three SNPs: rs6427553, rs7411035 and rs4656958. Limitations of the current study should be discussed.
5. Did the authors detect serological levels of biomarkers among different genotypes of rs6427553, rs7411035 and rs4656958 among the IS patients?
6. English language needs to be checked and polished.

·

Basic reporting

This paper introduces the association of single nucleotide polymorphisms in ITLN1 gene with ischemic stroke risk in the Chinese population. It is a novel study in the related field. The paper is reasonably designed with sufficient evidence and arguments, but it needs some improvement before it is accepted for publication. My specific comments are as follows:

Q1 In the introduction part of this article, should provide more background information, in order to understand the current research status of IS and genetic factors.

Q2 The reference for ischemic stroke is outdated, and it is recommended to replace it with data from the last 3 years.

Q3 This paper mainly studies the ITLN1 gene, but only serum ITLN1 is introduced in the introduction and discussion part of this paper. It is suggested to introduce the relevant information of ITLN1 gene.

Q4 In table 3 has a concise expression of the results, there is no need to list the data in the resulting statement in lines 105-159 of this paper.

Q5 The paper shows in lines 150 to 152 that the significant association between rs6427553, rs7411035 and rs4656958 and IS risk was not found in the subjects aged ≥60 years old and females. Is the statement in the conclusion too general?

Experimental design

Q6 This study included the population in Xi 'an area, can it represent the population of China?

Validity of the findings

Q7 This paper should discuss the limitations of this study in more detail and should discuss potential sources of bias and potential confounding factors that may affect the results.

·

Basic reporting

Shi and colleagues performed a case-control study of the relationship between the ITLN1 polymorphisms with Ischemic stroke in Chinese population well. This study encompasses some interesting points however several points should be addressed before acceptance for the publication. Comments and suggestions are below.
Generally, the manuscript could be improved with linguistic revision by a fluent English speaker. Some passages should be rephrased.

Experimental design

'no comment'

Validity of the findings

no comment

Additional comments

Abstract:
1) Add the numbers of the study groups and briefly explain the sanger sequencing and add the NM names of reference sequences in the Methods section as in the examples below.

Example 1)“We sequenced deoxyribonucleic acid from 96 patients with myelomeningocele. The 11 exons were amplified by polymerase chain reaction, and the products were sequenced with the Sanger method. Results were compared with reference sequences (NM_000454, NM_000636, and NM_001024466) obtained from University of California Santa Cruz Genome Browser. Observed alleles that differed from the reference sequences were considered novel variants.”
Example 2) “Methods: MMP8 rs1940475 and rs3765620, and MMP10 rs17860949 from 700 IS patients and 700 controls were genotyped by the MassARRAY iPLEX platform. The impact of polymorphisms on IS risk was evaluated by logistic regression analysis.”

2) The sentence “By comparing with the Sanger sequencing results and the reference sequence of the NCBI website” which is written in the result part, should be explained in methods.
3) “Logistic regression analysis showed that ITLN1 (rs6427553, rs7411035 and rs4656958) were associated with a decreased risk of IS, especially in people less than 60 years old and males” Who do you mean by men over sixty? Is it IS patient or Control group?
4) The sentence “In short, we observed a correlation between ITLN1 variants (rs6427553, rs7411035 and rs4656958) and IS risk in the Chinese subjects, laying the foundation for ITLN1 gene as a biomarker for ischemic stroke.” is not appropiate for the conclusion part of the study. Please state comment about this correlation.

Introduction:
1) The incidance of IS has been mentioned too much in the first paragraph. It can be shortened.
2) “ Intelectin-1” should be used instead of “Omentin-1” in all over manuscript.
3) Please state information about ITLN1 gene, its polymorphisms and the IS relationship.
4) “In the present study, we used Sanger sequencing to complete the PCR amplification of the ITLN1 gene promoter region, and then compared the obtained sequence with the sequence on NCBI to identify SNPs of the ITLN1 gene promoter region. In addition, the association of these loci with the risk of IS was analyzed using logistic regression.The results will lay a foundation for finding the diagnostic markers of IS”.
Please modify this paragraph to include the purpose of the study.

Materials & Methods:
1) The subsections “DNA extraction, Primer design and DNA amplification” and “Sanger sequencing of PCR products” should be combined under the subheading “Sanger sequencing of PCR products”.
2) The sentence “We used G*power 3.1.9.7 software to estimate the sample size of the case group and the control group through t-test. The parameter settings are: Tail=2, Effect size = 0.20, ³ )= 0.05, Power=0.945, Allocation ratio =1.” should be replaced in Statistical analyses.
3) Please specify in which program the primer design was designed.
4) Line 82-83. The sentences “The positive and negative primers were shown in the Online Resource 1” should be deleted.
5) Line 84. please state the full name of the Genewiz company in parentheses.
6) “According to the results of agarose gel electrophoresis, 4% L of the PCR target fragment amplification product with clear and complete bands was sent to Tsingke Biotechnology Co., Ltd. for two-way Sanger sequencing. After obtaining base sequence of the target fragment, it was compared with the reference sequence of ITLN1 gene promoter region provided by NCBI to determine whether there were single nucleotide polymorphisms in ITLN1 gene promoter region.”
The paragraph is very detailed and hard to follow. Authors should simplify the methods and specify the program the sanger sequencing was analyzed.
7) Line 96. This sentences “After the sequence alignment was completed, the SNP was obtained.” should be deleted.
8) “According to Fisher’s exact test, significant SNPs were screened in the control group and the case group (p <0.05). Then, chi-square test was used to perform Hardy-Weinberg equilibrium (HWE) test on significant sites to evaluate whether the enrolled samples were representative of the population.”
Instead of this explanation Authors may explain the method by using the sentence as below.
“ Genotype and allele distributions of detected SNPs were compared using Fisher’s exact test. The Hardy–Weinberg equilibrium and genotype distribution in case and control subjects were analyzed using the χ2 test.

9) Please specify the models used in the Genotype–phenotype associations, such as dominant and recessive model, in the statistical analysis section.
10) Please state the adjusted for confounding variables used in logictic regression analysis.
11) Please specify whether the age and gender classification were done in the whole group or among the case-control groups in table 4.
12) Please provide an explanation of the abbreviations FPRP and MDR.

Results:
1) In line 115, As shown in Table 1 should be written instead of table 1.
2) “As shown in the allele model, people with rs6427553-T, rs7411035-G and rs4656958-A had a lower risk of IS (OR : 0.84, 95%CI: 0.72-0.99 p= 0.032; OR : 0.82, 95%CI: 0.70-0.97, p= 0.017; OR : 0.76, 95%CI: 0.64-0.90 p= 0.002).”
In the sentence please identify the people whether they are IS patients or control group.
Also, in line 119 please state the word respectively in the end of the sentence.
3) In lin 121 please mention the general population whether they are IS patients or control group.
4) The lines of123-127 are also mentioned in the table 2, just p<0.05 can be specified.
5) Instead of the lines 128-133 as shown in table 3, the significant result should be mentioned.
6) “We did age and gender stratification analysis to assess the association between
polymorphisms and IS risk.”
In which group this stratification was done ? Authors should mention this.
7) In line 136 please specify the people.
6) Please state the number of case and control in figure 1 and tables.
7) Please delete the “A: minor alleles; B: major alleles.” in table 1.
9) Please delete the column of the gene in table 1.
10) Please delete the “logistic regression analysis with adjustments for age and gender.” in table 4.
Discussion:
1) In the discussion section, it is recommended that the article ** mentioned below be discussed and included in the sources.
2) The functional effect of ITLN1 variants (rs6427553, rs7411035 and rs4656958) should be discussed by performing in silico analysis.
3) The association between SNPs and IS should be discused more detailed.

---

## Round 0.2 · Minor Revisions

Thank you for thoroughly addressing the reviewers’ comments. All three original reviewers reviewed your resubmitted manuscript and two of them have minor comments that need to be addressed. One of the reviewers kindly provided an annotated copy of the sections of the manuscript and excerpts from your rebuttal that they wish to be addressed. First, your results section needs to be more concise and succinct. Additionally, a reference needs to be included and the last paragraph of your introduction needs to include the goal of your study.

Please, submit a detailed rebuttal which shows where and how you have taken all comments and suggestions into consideration. If you do not agree with some of the reviewers’ comments or suggestions, please explain why. Your rebuttal will be critical in making a final decision on your manuscript. Please, note also that your revised version may enter a new round of review by the same or by different reviewers. Therefore, I cannot guarantee that your manuscript will eventually be accepted.

Reviewer 1 ·

Basic reporting

No additional comments.

Experimental design

No additional comments.

Validity of the findings

No additional comments.

Additional comments

The results section is too wordy, especially for the too many comparison outcomes. Onceagain, this section should be concise.

·

Basic reporting

no comment

Experimental design

no comment

Validity of the findings

no comment

Additional comments

The association between single nucleotide polymorphisms in the ITLN1 gene and the risk of ischemic stroke in the Chinese population presented in this paper is a new study in the related field. The topic of the paper is novel and the experimental design is reasonable. After revision, the evidence is sufficient, the data provided are rigorous and reliable, and the arguments in the discussion section are clear and consistent with the topic. The overall quality of the article is high and it is recommended for publication.

·

Basic reporting

no comment

Experimental design

no comment

Validity of the findings

no comment

Additional comments

no comment

---

## Round 0.3 · accepted · Accept

Thank you for thoroughly addressing the reviewers' comments and thus greatly improving your manuscript.

Reviewer 1 ·

Basic reporting

No additional comments.

Experimental design

No additional comments.

Validity of the findings

No additional comments.

·

Basic reporting

No comment

Experimental design

No comment

Validity of the findings

No comment

Additional comments

No comment